# New-Onset Atrial Fibrillation in Acute Myocardial Infarction Is a Different Phenomenon than Other Pre-Existing Types of That Arrhythmia

**DOI:** 10.3390/jcm11154410

**Published:** 2022-07-28

**Authors:** Monika Raczkowska-Golanko, Krzysztof Młodziński, Grzegorz Raczak, Marcin Gruchała, Ludmiła Daniłowicz-Szymanowicz

**Affiliations:** 1Department of Cardiology and Electrotherapy, Medical University of Gdansk, 80-211 Gdańsk, Poland; mrg@gumed.edu.pl (M.R.-G.); krzysztofmlodzinski@gumed.edu.pl (K.M.); grzegorz.raczak@gumed.edu.pl (G.R.); 2First Department of Cardiology, Medical University of Gdańsk, 80-211 Gdańsk, Poland; marcin.gruchala@gumed.edu.pl

**Keywords:** atrial fibrillation, acute myocardial infarction, new-onset atrial fibrillation

## Abstract

(1) Background: Atrial fibrillation (AF) in acute myocardial infarction (AMI) could worsen the prognosis. Yet, there is no definitive answer to whether new-onset AF (NOAF) is a more aggravating diagnosis than other types of that arrhythmia. The purpose of our study was to compare in-hospital clinical course and outcomes of NOAF patients contrary to patients with other pre-existing types of AF. (2) Methods: AMI patients hospitalized in the high-volume cardiological center within 2017–2018 were included in the study. NOAF was noticed in 106 (11%) patients, 95 (10%) with an AF history and AF during AMI formed the AF group, 60 (6%) with an AF history but without AF during AMI constituted the Prior-AF group, and 693 (73%) patients were without an AF before and during AMI. Medical history, routinely monitored clinical parameters, and in-hospital outcomes were analyzed between the groups. (3) Results: NOAF patients, contrary to others, initially had the highest high-sensitivity troponin I (hsTnI), B-type natriuretic peptide (BNP), C-reactive protein (CRP), and glucose levels, and the lowest potassium concentration, with the worst profile of changes for that parameter within the first four days of hospitalization. NOAF patients had the highest rate of ST-elevated AMI (40%), the longest hospitalization (*p* < 0.001), and the highest in-hospital mortality (*p* < 0.001). Not NOAF, but other AF groups (AF and Prior-AF groups) were more burdened with the previous comorbidities. (4) Conclusions: NOAF could be a distinct phenomenon in AMI patients, identifying those with the worst clinical in-hospital course and outcomes as compared to other types of AF.

## 1. Introduction

Atrial fibrillation (AF) is the most common clinical arrhythmia, affecting 2–4% of the general population [1]. With an incidence of 5 to 23% [2,3,4,5] it is the most frequent arrhythmia connected with acute myocardial infarction (AMI). New-onset AF (NOAF), defined as newly diagnosed AF in AMI, constitutes a particular type of that arrhythmia. According to data from the literature [6,7,8] and the results of our previous study [9], NOAF is connected with worse clinical characteristics and poor outcomes in comparison to other patients with AMI [10]. However, there is no precise answer to whether NOAF is a more aggravating diagnosis in AMI patients than other types of AF. There were a few studies in the literature that tried to compare NOAF patients with other types of AF [10,11,12,13,14,15], however, they analyzed only selected groups of patients, preferably ST-elevation myocardial infarction (STEMI) [14,15], or only patients with AMI treated invasively [10], or only compared NOAF with one of other types of AF (i.e., chronic AF and NOAF [13], pre-existing AF and NOAF AF [12]), or were performed earlier, before the widespread availability of thrombolytic and percutaneous treatment for AMI patients [11]. As a result, comparing these data is challenging, and they differ significantly. The purpose of our study was to compare in-hospital clinical course and prognosis between NOAF patients and other pre-existing types of AF. We tried to answer the question of whether the NOAF is the same disease as pre-existing arrhythmia in AMI patients.

## 2. Materials and Methods

This study is a sub-analysis of our previous retrospective research, where the recruitment process was precisely described [9]. The study population consisted of consecutive AMI patients hospitalized in the University Clinical Centre of Gdansk from January 2017 to December 2018. The data was collected through MedStream Designer, which is fully integrated with the hospital information system. The exclusion criterion was age younger than 18 years. AMI diagnosis was based on the appropriate measures [16,17].

All patients were divided into four groups:NOAF (group of patients with any newly diagnosed AF that appeared during AMI hospitalization without a prior diagnosis of AF as it was precisely described [9]);AF (group of patients with a previously documented diagnosis of AF who additionally had AF during AMI hospitalization);Prior-AF (group of patients with a previously documented diagnosis of AF who had not developed AF during AMI hospitalization); andNon-AF (group of patients with no evidence of AF during AMI hospitalization and without the prior AF diagnosis).

For all patients, detailed medical history and clinical parameters, as well as in-hospital treatment and outcomes, were analyzed. Additionally, the course for laboratory parameters within the first four consecutive days of AMI hospitalization was taken into consideration. The pharmacotherapy at discharge (that was under the discretion of the attending physician) was thoroughly collected. The Independent Bioethical Committee approved the study’s protocol for Scientific Research of the Medical University of Gdansk (NBBN/290/2018). Due to the retrospective character of the study based on the routine clinical parameters, the necessity for written and informed consent was waived.

### Statistical Analysis

Continuous data are presented as median (25th–75th percentile), and categorical as numbers (n) and percentages (%). We performed the Shapiro–Wilk test to determine whether our data were normally distributed; most of the analyzed parameters did not have a normal data distribution, even after logarithmic transformation; therefore, we selected appropriate statistical analysis methods based on non-parametric tests. Comparisons between all groups were performed by the Kruskal–Wallis test for continuous variables (with Dunn’s post-hoc test for the multiple comparisons with Bonferroni adjusted) or by the chi-square test or Fisher test for categorical variables. The significance of differences for laboratory parameters analyzed within the first four consecutive days of AMI hospitalization was assessed using the Kruskal–Wallis tests for the group comparison and Friedman test and paired Wilcoxon test with Bonferroni adjusted. Linear mixed-effects models were used for data analysis with repeated measurements of the same variable for the four time points (from day 1 to day 4), to select the optimal set of predictors, the model was estimated using the backward stepwise method and Akaike Information Criterion. Values of *p* < 0.05 were considered significant. The statistical analysis was conducted with Statistics and R 4.0.5. environment (R Core Team, Vienna, Austria).

## 3. Results

### 3.1. Baseline Clinical Characteristics

As it was documented in our previous study [9], 954 patients with AMI were enrolled in the study. The NOAF group consisted of 106 patients (11%), whereas the AF group (a prehospital diagnosis of AF and AF during hospitalization) included 95 patients (10%), the Prior-AF group (patients with a previously documented diagnosis of AF who had not developed AF during hospitalization) - 60 patients (6%), and the remaining 693 (73%) were patients without AF (Non-AF group). Table 1 presents the baseline clinical characteristics of all studied patients. Patients with any AF (including the NOAF group) were older than Non-AF patients. Regarding comorbidities, AF and Prior-AF patients were more burdened with diseases; interestingly, the NOAF group was similar to Non-AF patients in this issue, with the only exception being in the rate of prior stroke. In the analysis of the prehospital pharmacological treatment, it was easy to notice the better treatment with angiotensin-converting enzyme (ACE) inhibitors/sartans and statins in both groups of patients with a previous history of AF contrary to NOAF and Non-AF patients.

### 3.2. In-Hospital Characteristics and Outcomes

Among all analyzed groups, NOAF patients had the highest rate of STEMI-40%, more than two-fold higher than in other patients with AF (AF and Prior-AF groups). Almost 100% of enrolled patients had coronary angiography during the hospitalization, and 82% had a percutaneous coronary intervention (PCI), as is presented in Table 2. NOAF patients had the worst in-hospital prognosis, including the highest rate of adverse events (malignant arrhythmias or stroke) and in-hospital mortality: twice more than in the AF group and four to six times more than the remaining groups (Table 2). The majority of NOAF patients (85%), in contrast to the AF group (36%), had sinus rhythm at discharge.

### 3.3. In-Hospital Laboratory and Echocardiographic Parameters

Table 3 presents the results of the laboratory parameters measured on the first day of hospitalization, and, additionally, the maximal level of high sensitivity troponin I (hsTnI) and echocardiographic measures. The four analyzed groups significantly differ regarding those parameters: most of which (B-type natriuretic peptide (BNP), troponin, C-reactive protein (CRP), glucose, and hemoglobin) were worse in the patients with any AF (NOAF, Prior-AF, AF groups) in comparison to the Non-AF group. The NOAF group was characterized by the highest level of hsTnI, BNP, CRP, and glucose, and the lowest potassium concentration. Total cholesterol (TC) and low-density lipoprotein cholesterol (LDL-C) were significantly higher in the Non-AF group in comparison to all AF patients. Patients with AF (NOAF, AF, and Prior-AF groups) had significantly worse left ventricular ejection fraction (LVEF), with the lowest level for the NOAF group. Patients without AF, as expected, had the smallest left atrium (LA) size. In patients with AF in AMI (NOAF and AF groups), the right ventricular size was the largest (Table 3).

### 3.4. In-Hospital Laboratory Parameters Dynamic

Figure 1, Figure 2, Figure 3, Figure 4 and Figure 5 present the dynamic changes in some of the laboratory parameters within the first four consecutive days of hospitalization. As is easy to note, the NOAF patients are characterized by the most prominent changes in CRP, leucocytes, and hsTnI, as well as the lowest potassium level. NOAF patients had the most significant increase in hsTnI level, with the maximum level being on the second day of hospitalization (Figure 1). Similarly, the NOAF patients had the highest CRP level, which steadily increased during the four days and was two to three times higher than in the Non-AF group (Figure 2). NOAF patients had the maximal values of leucocytes on the first day of hospitalization, with a peak on the second day, whereas the remaining AF patients (AF and Prior-AF groups) experienced a slight decrease of this parameter during the consecutive four days. (Figure 3). The potassium level was the lowest in the NOAF group throughout the whole four-day period of measurements (Figure 4). NOAF patients were characterized by the biggest reduction in hemoglobin level during the consequent four days of our observation (Figure 5).

In the linear mixed model analysis, the impact of some clinical characteristics was determined: age (*p* < 0.001), male sex (*p* = 0.008), and history of diabetes mellitus (*p* = 0.009) on CRP level, hypertension history (*p* = 0.017) on leucocytes level, male sex (*p* = 0.019), and prior MI (*p* = 0.008) on potassium level, age (*p* < 0.001), myocardial infarction history (*p* = 0.009), diabetes mellitus (*p* = 0.004), and history of stroke (*p* = 0.021) on hemoglobin level. However, no interactions with the assessment of the group effect and time effect were noticed.

### 3.5. Pharmacological Treatment at Discharge

The studied groups significantly differed in pharmacological treatment at discharge (Table 4). Patients with arrhythmia within AMI hospitalization (NOAF and AF groups) had prescribed NOACs and triple antithrombotic therapy more often than patients without AF onset during hospitalization (Prior-AF and Non-AF groups). Moreover, triple therapy was prescribed more often for AF group patients (70%) than for NOAF (57%).

## 4. Discussion

The main finding of our study is that NOAF is a distinct phenomenon in comparison to other pre-existing AF types in patients with AMI. The appearance of NOAF seems to be the indicator of poor AMI course and worse in-hospital prognosis contrary to patients with a previous history of AF, who are, however, more burdened with comorbidities before AMI, but had better prognosis within hospitalization. To the best of our knowledge, this is the first study in which the complex evaluation of routinely measured clinical and laboratory parameters regarding different types of AF in the modernly treated AMI patients, with special attention to NOAF patients, was performed.

Data from the literature confirms that AF is common in patients with AMI [2,10,11,12,13,14,18,19]. Our results are in line with that: every fifth enrolled patient (21%) had AF with the highest frequency for NOAF (11%). All patients with a history of AF (AF and Prior-AF groups) accounted for 16% (10% and 6% respectively). According to data from the literature, the clinical profile of patients with AF during AMI differs significantly from other patients: they are older, have an increased burden of cardiovascular risk factors like coronary artery disease, history of hypertension, diabetes mellitus, and stroke [12,20]. In our study, that characteristic was related only to patients with a prior history of AF, but not to NOAF. Contrary to possible predictions, patients with NOAF had a better medical history but were characterized by worse in-hospital clinical course and prognosis.

Our previous study proved that age, BNP, CRP, and LVEF were associated with NOAF [9]. One of the results of current research is that NOAF patients were characterized by the highest troponin level (Figure 1). Troponin concentration is a well-known measure of AMI intensity, with a high level in STEMI rather than NSTEMI [21], and according to some data, a prognostic factor of poor prognosis [22]. Our results confirm that information: NOAF patients had the highest rate of STEMI (40%), contrary to only 17% in AF and 15% in Prior-AF groups. Interestingly, Non-AF patients had a similar rate of STEMI (36%), however, their troponin level was slightly lower than in NOAF, but higher than in AF and Prior-AF groups (Table 3). We could suppose that NOAF presentation could be a consequence of severe myocardial necrosis and occur especially in sick patients with a large myocardial infarction. As NOAF patients have a more significant occurrence of STEMI, that can explain their worse outcomes [23]. In our study, NOAF patients had the highest in-hospital mortality (19%) in comparison to other sub-groups: twice that of AF patients (9%) and four to six times that of the other groups.

Regarding laboratory parameters, we noticed some important differences between the studied sub-groups in our study. Patients with AF (NOAF, AF and Prior-AF groups) had higher than Non-AF patients BNP level (Table 3). According to the literature, BNP could be elevated in patients with AF, and this elevation returns to normal value after sinus rhythm restoration, suggesting that BNP may play a role in predicting AF recurrence [24,25]. That could explain the above-mentioned differences. However, in our results, NOAF patients had the highest BNP level, which could suggest that NOAF is connected with the most prominent hemodynamic changes, in contrast with other AF during AMI. Our previous study demonstrated that BNP with a cut-off value of ≥340 pg/mL is a robust and independent predictor of NOAF [9]; that could suppose that the occurrence of AF itself is associated with higher BNP level. Inflammation, which can cause structural and electrical changes in the atrium, predisposing patients to AF, could be connected with the changes in the laboratory parameters such as CRP and white blood cells (WBC) [26,27]. For instance, CRP has been reported as a risk factor for AF episodes, including AF recurrences after successful cardioversion [28]. Similarly, WBC is one of the predictors of AF after cardiac surgery [29,30]. According to Yoshizaki et al., CRP and WBC were linked to NOAF in the early stages of STEMI, and an increase of both of those parameters was observed during the next days of hospitalization for AF patients [31]. Our results are in line with the above-mentioned information, and we revealed that patients with any AF in AMI (NOAF and AF groups) had two to three folds higher CRP levels compared with patients without AF (Table 3 and Figure 2). NOAF and AF groups had the highest WBC on admission, and the NOAF group had a peak WBC during the second day of in-hospital treatment (Table 3, Figure 3). Low serum potassium level is the next well-known characteristic linked to the development of AF in the general population [23,32,33,34]. In our previous study, potassium levels below 4.2 mmol/L were found to be crucial in revealing the NOAF probability [9]. The present study shows similar results, indicating the main difference between the compared groups on the first day of hospitalization: the lowest level was found in the NOAF group; AF and Prior-AF patients had higher potassium levels than NOAF, and the highest level was observed in the Non-AF group (Figure 4). Beginning from the second day of hospitalization, there were more differences: Prior-AF patients had higher potassium level than other groups, and NOAF patients always had the worst levels. That could be explained by the fact that in usual clinical practice, the patients with a documented history of AF usually receive more potassium supplements to prevent AF onset. Decreased hemoglobin level has been linked to poor outcomes in patients with AF in AMI [35]. In the presented study, NOAF patients had the most profound reduction in hemoglobin levels during the first four days of hospitalization (Figure 5). The highest level of hemoglobin was found in Non-AF patients.

Regarding echocardiographic parameters, the probability of AF increases with the enlargement of LA and reduction in LVEF [36,37]. Our latest study proved that LA diameter ≥ 41 mm and LVEF ≤ 44% were significant predictors of NOAF in the univariate analysis, with maintained significance for LVEF in the multivariate calculations [9], which is also in line with the latest research considering NOAF patients [38]. The present study shows that NOAF patients had the lowest LVEF (Table 3); however, the largest LA was not connected with the NOAF patients, but with patients from AF groups (patients with a previous history of AF and AF during AMI hospitalization). That result could confirm our supposition that NOAF is not a typical burden, but a consequence of severe AMI course.

When describing the pharmacological treatment of the studied patients, it should be mentioned that physicians make difficult therapeutic decisions when managing AF during the AMI, especially if AF onset is only during the acute phase of AMI, balancing embolic and hemorrhagic risks. These decisions are frequently based on expert consensus [38] and current guidelines [16,17,39]. Our study shows, however, high rate of recommended triple antithrombotic treatment (oral anticoagulation and dual antiplatelet therapy), but possibly not efficient: 70% for AF patients, 57% for NOAF and 45% for Prior-AF group (Table 4). Importantly, this is still a rate that is much higher than previously reported in other research [12,14], showing the growing awareness of the present recommendations.

## 5. Limitations

Our study presents some limitations. This single-center, retrospective study limits some of the data and parameters available in patients’ medical records. Due to the general nature of data encoding, only patients with AMI were accurately coded; therefore, we were unable to use the term acute coronary syndrome (we do not have any data about patients with unstable angina). Due to the retrospective nature of our research and the use of an anonymous medical database (MedStream Designer), we could not perform the adequate long-term follow-up. Another limitation is possibly overestimating the NOAF (qualifying here patients with previously undetected paroxysmal AF). On the other hand, we could have underestimated the proper frequency of AF (due to silent AF episodes). Moreover, patients with permanent AF and a history of AF and an episode of AF in AMI are in one group, which is a rather inhomogeneous group, but it was impossible to separate them correctly in a retrospective evaluation.

## 6. Conclusions

New-onset atrial fibrillation in acute myocardial infarction is a different phenomenon than other pre-existing types of that arrhythmia. Its appearance seems to be the indicator of poor AMI course and worse in-hospital prognosis contrary to patients with a previous history of AF, who are, however, more burdened with comorbidities before AMI, but had better prognosis within hospitalization.

## Figures and Tables

**Figure 1 jcm-11-04410-f001:**
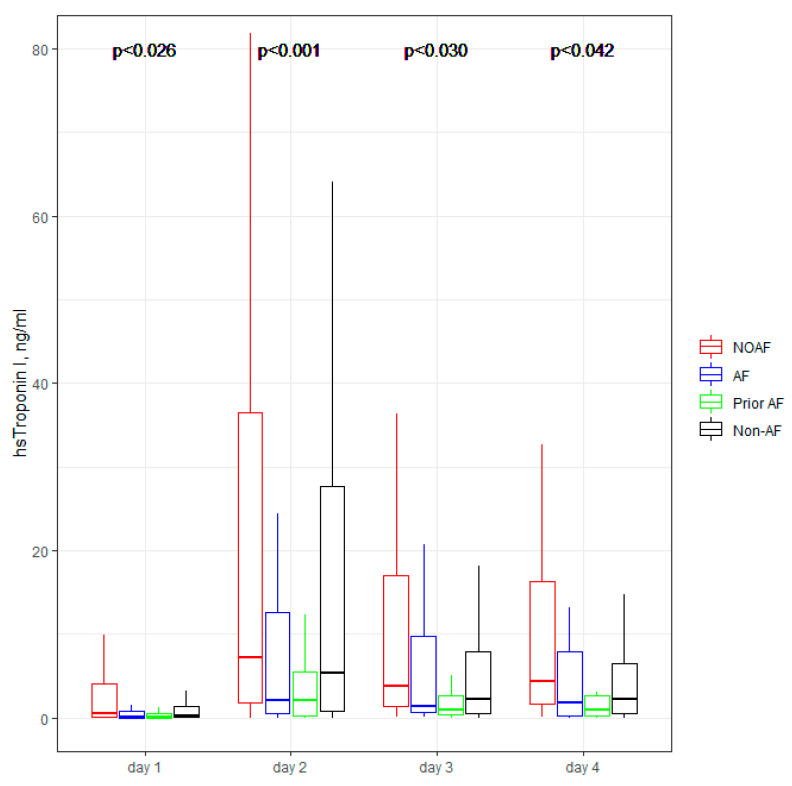
**hsTnI concentration within the first four days of hospitalization.** The center represents the median value. The upper and lower quartiles values are displayed with whiskers. *p*-values on the figure represent the group changes (the Kruskal–Wallis test). The time changes were calculated by the Friedman test and paired samples Wilcoxon test with Bonferroni adjusted (*p* < 0.001). hsTnI—high sensitivity troponin I; NOAF—group of patients with any newly diagnosed AF that appeared during AMI hospitalization; AF—group of patients with a previously documented diagnosis of AF who additionally had AF during AMI hospitalization; Prior AF—group of patients with a previously documented diagnosis of AF who had not developed AF during AMI hospitalization; Non-AF—group of patients with no evidence of AF during AMI hospitalization and without the prior AF diagnosis.

**Figure 2 jcm-11-04410-f002:**
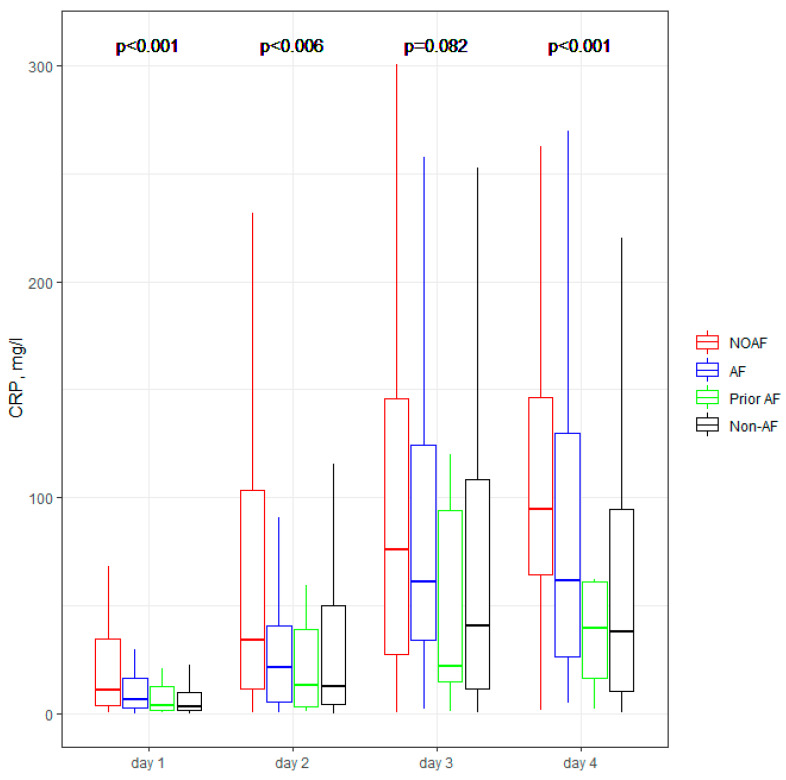
**CRP concentration within the first four days of hospitalization.** The center represents the median value. The upper and lower quartiles values are displayed with whiskers. *p*-values on the figure represent the group changes (the Kruskal–Wallis test). The time changes were calculated by the Friedman test and paired samples Wilcoxon test with Bonferroni adjusted (*p* < 0.001). CRP—C-reactive protein; NOAF—group of patients with any newly diagnosed AF that appeared during AMI hospitalization; AF—group of patients with a previously documented diagnosis of AF who additionally had AF during AMI hospitalization; Prior AF—group of patients with a previously documented diagnosis of AF who had not developed AF during AMI hospitalization; Non-AF—group of patients with no evidence of AF during AMI hospitalization and without the prior AF diagnosis.

**Figure 3 jcm-11-04410-f003:**
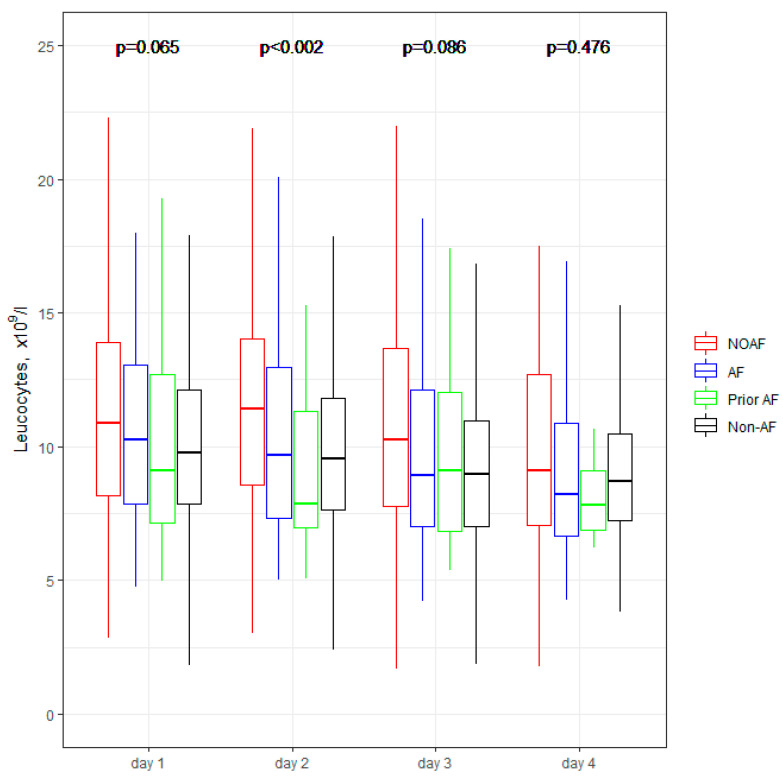
**Leucocytes within the first four days of hospitalization.** The center represents the median value. The upper and lower quartiles values are displayed with whiskers. *p*-values on the figure represent the group changes (the Kruskal–Wallis test). The time changes were calculated by the Friedman test and paired samples Wilcoxon test with Bonferroni adjusted (*p* < 0.001). NOAF—group of patients with any newly diagnosed AF that appeared during AMI hospitalization; AF—group of patients with a previously documented diagnosis of AF who additionally had AF during AMI hospitalization; Prior AF—group of patients with a previously documented diagnosis of AF who had not developed AF during AMI hospitalization; Non-AF—group of patients with no evidence of AF during AMI hospitalization and without the prior AF diagnosis.

**Figure 4 jcm-11-04410-f004:**
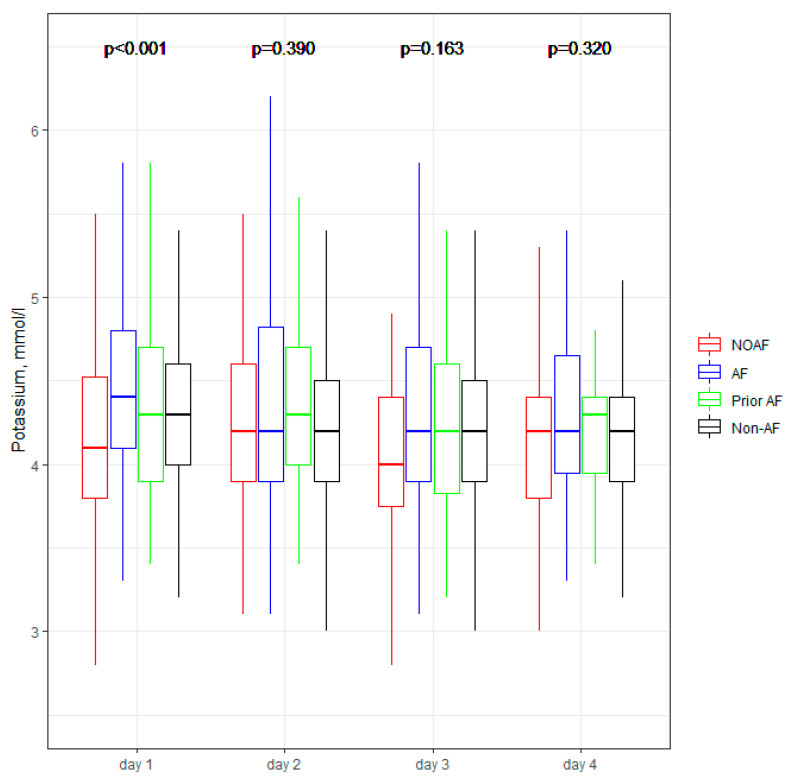
**Potassium concentration within the first four days of hospitalization.** The center represents the median value. The upper and lower quartiles values are displayed with whiskers. *p*-values on the figure represent the group changes (the Kruskal–Wallis test). The time changes were calculated by the Friedman test and paired samples Wilcoxon test with Bonferroni adjusted (*p* < 0.004). NOAF—group of patients with any newly diagnosed AF that appeared during AMI hospitalization; AF—group of patients with a previously documented diagnosis of AF who additionally had AF during AMI hospitalization; Prior AF—group of patients with a previously documented diagnosis of AF who had not developed AF during AMI hospitalization; Non-AF—group of patients with no evidence of AF during AMI hospitalization and without the prior AF diagnosis.

**Figure 5 jcm-11-04410-f005:**
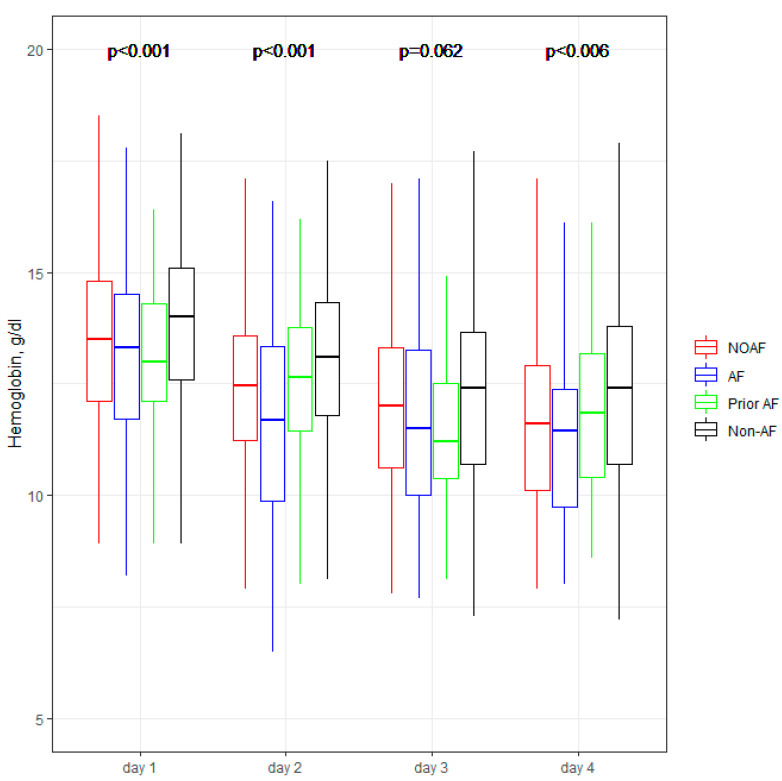
**Hemoglobin within the first four days of hospitalization.** The center represents the median value. The upper and lower quartiles values are displayed with whiskers. *p*-values on the figure represent the group changes (the Kruskal–Wallis test). The time changes were calculated by the Friedman test and paired samples Wilcoxon test with Bonferroni adjusted (*p* < 0.001). NOAF—group of patients with any newly diagnosed AF that appeared during AMI hospitalization; AF—group of patients with a previously documented diagnosis of AF who additionally had AF during AMI hospitalization; Prior AF—group of patients with a previously documented diagnosis of AF who had not developed AF during AMI hospitalization; Non-AF—group of patients with no evidence of AF during AMI hospitalization and without the prior AF diagnosis.

**Table 1 jcm-11-04410-t001:** Baseline clinical characteristics.

	NOAF * *n* = 106	AF § *n* = 95	Prior-AF ¶ *n* = 60	Non-AF *n* = 693	*p*
Age (years old)	73 (66–84)	74 (67–82)	72 (69–78)	65 (59–73), *,¶	0.001
Male sex, *n* (%)	67 (63%)	55 (58%)	42 (70%)	473 (68%)	0.172
Prior MI, *n* (%)	31 (29%), ¶	41 (43%)	32 (53%), *	172 (25%), §,¶	0.001
Prior revascularization (PCI/CABG), *n* (%)	26 (25%), §,¶	41 (43%), *	28 (47%), *	175 (25%), §,¶	0.001
Hypertension, *n* (%)	79 (75%), ¶	82 (86%)	55 (92%), *	502 (73%), §,¶	0.001
Diabetes mellitus, *n* (%)	31 (29%), §,¶	45 (47%), *	28 (47%), *	210 (30%), §,¶	0.001
Previous stroke, *n* (%)	10 (9.4%)	19 (20%)	5 (8%)	36 (5%), §	0.001
** *On-Admission Treatment* **					
Aspirin, *n* (%)	43 (41%)	32 (34%)	22 (37%)	259 (38%)	0.826
ACE inhibitors/sartans, *n* (%)	53 (50%), §,¶	67 (71%), *	44 (73%), *	346 (50%), §,¶	0.001
Statins, *n* (%)	41 (39%), ¶	48 (51%)	37 (62%), *	249 (36%), §,¶	0.001

Abbreviations: *p*-value: for differences among all groups with Kruskal–Wallis test for continuous variables or with chi-square test for categorical variables, *p* < 0.05 in post-hoc tests for differences with group NOAF (*), AF (§), or Prior-AF (¶). ACE—angiotensin-converting enzyme; BMI-body max index; CABG—coronary artery bypass grafting; ICD—implantable cardioverter-defibrillator; MI—myocardial infarction; PCI—percutaneous coronary intervention.

**Table 2 jcm-11-04410-t002:** Types of AMI, results of coronary angiography, and in-hospital prognosis.

	NOAF * *n* = 106	AF § *n* = 95	Prior-AF ¶ *n* = 60	Non-AF *n* = 693	*p*
** *Types of Myocardial Infarction* **					
ST-elevation MI, *n* (%)	42 (40%), §,¶	16 (17%), *	9 (15%), *	260 (36%), §,¶	0.001
Non-ST-elevation MI, *n* (%)	64 (60%), §,¶	79 (83%), *	51 (85%), *	423 (62%), §,¶	0.001
In-hospital coronary angiography, *n* (%)	99 (93%)	90 (95%)	58 (97%)	674 (97%)	0.121
In-hospital PCI, *n* (%)	81 (76%)	69 (73%)	49 (82%)	580 (83%)	0.413
** *In-Hospital Prognosis* **					
Length of hospitalization (days)	10 (7–17), ¶	9 (6–14), ¶	7 (5–10), *, §	6 (5–8), *,§	0.001
VT during hospitalization, *n* (%)	6 (6%)	3 (3%)	2 (3%)	15 (2%)	0.166
VF during hospitalization, *n* (%)	14 (13%)	4 (4%)	1 (2%)	46 (7%)	0.023
AVB III during hospitalization, *n* (%)	6 (6%)	1 (1%)	1 (2%)	7 (1%), *	0.013
Stroke during hospitalization, *n* (%)	3 (3%)	2 (2%)	1 (2%)	3 (0.43%)	0.023
In-hospital mortality, *n* (%)	19 (18%), ¶	9 (9%)	2 (3%), *	28 (4%), *	0.001
Sinus rhythm at discharge, *n* (%)	74 (85%), §,¶	31 (36%), *,¶	52 (89%), *,§	661 (99%), *,§	0.001

Abbreviations: *p*-value: for differences among all groups with Kruskal–Wallis test for continuous variables or with chi-square test for categorical variables, *p* < 0.05 in post-hoc tests for differences with group NOAF (*), AF (§), or Prior-AF (¶). AVB—atrioventricular block; MI—myocardial infarction; PCI—Percutaneous coronary intervention; SR—sinus rhythm; VF—ventricular fibrillation; VT—ventricular tachycardia.

**Table 3 jcm-11-04410-t003:** Laboratory and echocardiographic parameters of the studied groups.

	NOAF * *n* = 106	AF § *n* = 95	Prior-AF ¶ *n* = 60	Non-AF *n* = 693	*p*
BNP, pg/mL	491 (193–1087), ¶	270 (158–895)	248 (78–622), *	114 (43–362), *, §	0.001
hsTnI, ng/mL	0.64 (0.06–4.84), §,¶	0.148 (0.04–0.78), *	0.127 (0.03–0.55), *	0.215 (0.05–1.40)	0.026
hsTnI max, ng/mL	10.59 (2.98–36.62), §,¶	3.11 (0.91–13.48), *	2.37 (0.78–6.64), *	6.51 (1.35–28.11), *,§,¶	0.001
CK-MB, ng/mL	4.75 (2.2–14)	4 (2.0–7.5)	3.35 (1.5–6.2)	4.05 (2.1–11.2)	0.136
CRP, mg/L	11.2 (3.55–34.5), ¶	6.5 (2.8–16.6), *	3.56 (1.8–12.4), *	3.4 (1.4–9.9 ), *,§	0.001
Sodium, mmol/L	138 (135–140)	138 (135–140)	138 (136–140)	138 (136–140)	0.167
Potassium, mmol/L	4.1 (3.8–4.5), §	4.4 (4.1–4.8), *	4.3 (3.9–4.7)	4.3 (4.0–4.6), *	0.007
Hemoglobin, g/dL	13.5 (12.1–14.8)	13.3 (11.7–14.5)	13.0 (12.1–14.3)	14 (12.6–15.1), §,¶	0.001
Leucocytes, × 10^9^/L	10.87 (8.18–13.91)	10.23 (7.84–13.3)	9.08 (7.13–12.69)	9.77 (7.86–12.12), *	0.065
Neutrophil to lymphocyte ratio	3.81 (2.2–6.8)	3.82 (2.5–8.0)	3.8 (2.5–6.5)	3.08 (2.0–5.1), §	0.002
Total cholesterol, mg/dL	169 (129–191)	148 (128–189)	159 (136–196)	181 (148–218), *,§,¶	0.001
LDL-C, mg/dL	98 (64–124)	87 (72–107)	94 (78–130)	109 (80–145), *,§	0.001
Creatinine, ml/dL	0.96 (0.78–1.24), §	1.14 (0.94–1.48), *	0.95 (0.8–1.33)	0.92 (0.78–1.13), §	0.004
TSH, uU/L	1.16 (0.66–1.85)	1.13 (0.60–2.22)	1.24 (0.82–2.59)	1.06 (0.48–1.67)	0.147
FT3, pmol/L	2.97 (2.75–3.30)	3.41 (2.54–3.78)	2.94 (2.54–3.42)	3.12 (2.67–3.7)	0.696
FT4, pmol/L	14.74 (13.40–16.10)	14.07 (12.36–15.27)	13.44 (12.49–14.54)	12.71 (11.31–14.51), *	0.009
Glucose, mg/dL	155 (120–219), §,¶	132 (101–186), *	118 (106–163), *	126 (103–172), *	0.001
** *Echocardiographic Parameters* **					
LVEF, %	40 (33–50)	44 (32–55)	49 (40–55)	50 (41–58), *,§	0.001
LA size, mm	42 (38–46), §	45 (41–50), *,¶	42 (38–45), §	39 (35–42), *,§,¶	0.001
LVIDd, mm	50 (44–55)	50 (46–56)	49 (45–56)	49 (45–53)	0.204
RVID, mm	42 (34–44)	42 (35–49)	36 (32–43)	35 (32–39), *,§	0.001
TAPSE, mm	19 (15–22)	17 (15–20)	19 (17–22)	21 (18–24), *,§	0.001
RVSP, mmHg	43 (35–47)	45 (35–46)	40 (31–47)	40 (30–46)	0.410

Abbreviations: *p*-value: for differences among all groups with Kruskal–Wallis test for continuous variables or with chi-square test for categorical variables, *p* < 0.05 in post-hoc tests for differences with group NOAF (*), AF (§), or Prior-AF (¶). BNP—B-type natriuretic peptide; CK-MB—creatine kinase muscle-brain; CRP—C-reactive protein; FT3-free triiodothyronine; FT4—free thyroxine; hsTnI—high sensitivity troponin I; LA—left atrium; LDL—C-low-density lipoprotein cholesterol; LVIDd—left ventricular internal diameter end diastole; LVEF-left ventricular ejection fraction; RVIDd—right ventricular internal dimension; RVSP-right ventricular systolic pressure; TAPSE—tricuspid annular plane systolic excursion; TSH—thyroid-stimulating hormone.

**Table 4 jcm-11-04410-t004:** Pharmacological treatment at discharge.

	NOAF *n*= 86	AF *n* = 85	Prior-AF *n* = 58	Non-AF *n* = 662	*p*
Beta-blockers, *n* (%)	76 (88%)	76 (89%)	48 (83%)	575 (87%)	0.677
ACE inhibitors/sartans, *n* (%)	73 (85%)	73 (86%)	47 (81%)	608 (92%)	0.006
Statins, *n* (%)	81 (94%)	74 (87%)	53 (91%)	633 (96%)	0.011
** *Antithrombotic Therapy* **					
Aspirin, *n* (%)	76 (88%)	74 (87%)	53 (91%)	639 (97%)	0.001
Clopidogrel, *n* (%)	72 (84%)	72 (85%)	51 (88%)	495 (75%)	0.012
Ticagrelor, *n* (%)	3 (3%)	0 (0%)	0 (0%)	145 (22%)	0.001
Vitamin K antagonists, *n* (%)	8 (9%)	24 (28%)	10 (17%)	13 (2%)	0.001
NOACs, *n* (%)	54 (63%)	51 (60%)	23 (40%)	12 (2%)	0.001
Low-molecular-weight heparins, *n* (%)	7 (8%)	8 (10%)	9 (16%)	18 (3%)	0.001
** *Triple Antithrombotic Therapy* **					
Aspirin + Clopidogrel + Vitamin K antagonists	8 (9%)	19 (22%)	10 (17%)	9 (1%)	0.001
Aspirin + Clopidogrel + NOACs	40 (47%)	40 (47%)	16 (28%)	12 (2%)	0.001
Aspirin + Clopidogrel + LMWH	1 (1%)	1 (1%)	0 (0%)	1 (1%)	0.001
** *Double Antithrombotic Therapy* **					
Aspirin + Clopidogrel	14 (16%)	3 (4%)	16 (28%)	457 (69%)	0.001
Aspirin + Ticagrelor	2 (2%)	0 (0%)	0 (0%)	137 (21%)	0.001

ACE—angiotensin-converting enzyme; ARBs—angiotensin receptor blockers; LMWH—low-molecular-weight heparin; NOACs—novel oral anticoagulants.

## Data Availability

Data are available on request due to privacy and ethical restrictions.

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
