# Peer review of "New-Onset Atrial Fibrillation in Acute Myocardial Infarction Is a Different Phenomenon than Other Pre-Existing Types of That Arrhythmia"

_jcm, 2022, doi:10.3390/jcm11154410_

Round 1

Reviewer 1 Report

This article presents a retrospective study of whether new onset AF during hospitalization for acute MI is a marker of more severe form of the disease with worse hospital outcomes than other variants of AF. The authors analyzed the available information from hospital records of the patients treated for acute MI and effectively demonstrated the significantly higher levels of Troponine, BNP, CRP followed by significantly higher rates of in-hospital ventricular arrhythmias and mortality in the patients with the new onset AF compared to those with MI and previously documented AF. In general, these data indicate that the new onset AF may have the separate clinical significance in the evaluation of patients with acute MI. No doubt, this finding is of interest to all involved in the medical care of these patients .

Comments:

The Table 1 shows the significant differences between groups in “Baseline clinical characteristics”, e.g. in age. However, there is no mention of any adjustment for that difference in the between-group comparison of the lab markers/clinical events or any other way to address the possible confounding effect/bias.

Also, regarding the Table 1 - it presents the comparison of 4 groups assessed with only one p-value, however, no methods of multiple comparisons is mentioned in the Methods section. If the p-values in the last column refer for the pairwise group comparison then the respective groups should be clearly indicated. No doubt, multiple comparison like ANOVA followed by the pairwise single comparisons of the NOAF to other groups (as well as the comparison of AF with prior AF) is the standard/appropriated approach in this case.

The same is true for the analysis of the serial lab data during first four days. There are the groups and repeated measures, which normally prompts the analysis of the mixed model (e.g. ANOVA or non-parametric analog) with the assessment of the group effect, time effect, and the interaction between them. Currently, the p-values presented on the graphs seem to assess the time effect only (of course, the p-values should be explained in the figure’s legend as well as what is displayed by the figures, e.g. The center represents the median/mean value. The maximum and minimum/upper and lower quartiles/SD/SE values are displayed with whiskers).

Reviewer 2 Report

Monika Raczkowsa-Golanko performed a retrospective analysis of patients admitted to the hospital with acute coronary syndrome, stratified by atrial fibrillation (AF): new onset-AF (NOAF), prior AF, prior and current AF, and no-AF. They found a higher prevalence of STEMI, lower LVEF, higher troponin and BNP levels, as well as worse outcome in NOAF compared to remaining patients.

The manuscript is well-written. Concerning statistics, I do not know why the authors did not perform testing for normal distribution and parametric testing in normally distributed variables. Furthermore, correction for multiple testing (such as Bonferroni) may be necessary in Table 3, for example. In my opinion, the bad results of NOAF patients are quite biased, because NOAF seems to occur especially in sick patients with a large myocardial infarction. To adequately assess the effect of NOAF on outcome, the authors should adjust for other predictors of worse outcomes, for example with multiple regression analysis.

Figures: Boxplots may be better in showing the differences between groups.

References are adequate.

Do the authors have any information on long-term outcome or rehospitalization rate?

The paper of Halima et al may be included or discussed (10.1016/j.acvdsp.2020.10.341; no COI from my side)
